# Pregnancy of unknown location: external validation of the hCG-based M6NP and M4 prediction models in an emergency gynaecology unit

Johan Fistouris [1,2] Christina Bergh,[1,2] Annika Strandell[1,2]

[1]Department of Obstetrics and Gynecology, Institute of Clinical Sciences, Sahlgrenska Academy, University of Gothenburg, Goteborg, Sweden
[2]Region Västra Götaland, Department of Gynecology and Reproductive Medicine, Sahlgrenska University Hospital, Gothenburg, Sweden

**Correspondence to**
Dr Johan Fistouris;
johan.fistouris@vgregion.se

## ABSTRACT

**Objective** To investigate if M6NP predicting ectopic pregnancy (EP) among women with pregnancy of unknown location (PUL) is valid in an emergency gynaecology setting and comparing it with its predecessor M4.

**Design** Retrospective cohort study.

**Setting** University Hospital.

**Participants** Women with PUL.

**Methods** All consecutive women with a PUL during a study period of 3 years were screened for inclusion. Risk prediction of an EP was based on two serum human chorionic gonadotropin (hCG) levels taken at least 24 hours and no longer than 72 hours apart.

**Main outcome measures** The area under the ROC curve (AUC) expressed the ability of a model to distinguish an EP from a non-EP (discrimination). Calibration assessed the agreement between the predicted risk of an EP and the true risk (proportion) of EP. The proportion of EPs and non-EPs classified as high risk assessed the model's sensitivity and false positive rate (FPR). The proportion of non-EPs among women classified as low risk was the model's negative predictive value (NPV). The clinical utility of a model was evaluated with decision curve analysis.

**Results** 1061 women were included in the study, of which 238 (22%) had a final diagnosis of EP. The AUC for EP was 0.85 for M6NP and 0.81 for M4. M6NP made accurate risk predictions of EP up to predictions of 20% but thereafter risks were underestimated. M4 was poorly calibrated up to risk predictions of 40%. With a 5% threshold for high risk classification the sensitivity for EP was 95% for M6NP, the FPR 50% and NPV 97%. M6NP had higher sensitivity and NPV than M4 but also a higher FPR. M6NP had utility at all thresholds as opposed to M4 that had no utility at thresholds≤5%.

**Conclusions** M6NP had better predictive performance than M4 and is valid in women with PUL attending an emergency gynaecology unit. Our results can encourage implementation of M6NP in related yet untested clinical settings to effectively support clinical decision-making.

## INTRODUCTION

To reduce maternal complications from ectopic pregnancy (EP), an early diagnosis using transvaginal ultrasound (TVUS) is essential.[1] Between 30% and 40% of EP remain undiagnosed after the first examination of

## STRENGTHS AND LIMITATIONS OF THIS STUDY

⇒ This is the first study to test the transportability (generalisability) of the M6NP prediction model to an emergency gynaecology setting, originally developed for early pregnancy assessment units in the UK.

⇒ The rate of ectopic pregnancy was twice as high in comparison with the original M6NP cohort and we also included inpatients contributing to a different case mix.

⇒ A limitation was the retrospective study design, although common for validation studies.

⇒ Another limitation was the complete case analysis; however a sensitivity analysis adding all women regardless of time interval between two hCG showed consistent performance for M6NP.

women presenting with pain or bleeding in early pregnancy and will be managed as a pregnancy of unknown location (PUL) until a final diagnosis is established.[2] The reported rupture rate of EP among women with a PUL range from 0.03% to 2.4% and sometimes necessitates blood transfusion.[3–6] A key concept during follow-up is timely diagnosis and treatment of an EP while avoiding unnecessary visits for the remaining majority of women with PUL that eventually will be diagnosed with an intrauterine pregnancy (IUP) or a spontaneously resolving PUL (failed PUL).[7] The change in serum human chorionic gonadotropin (hCG) levels between two samples measured 48 hours apart indicates if a woman is at high or low risk of having an EP and decisive of planning follow-up visits.[8] In a systematic review and meta-analysis the widely reported logistic regression model 'M4' that is based on two hCGs had the best performance predicting the presence of an EP among women with PUL.[9–12]

M4 was developed in an early pregnancy assessment unit (EPAU) in London and has mostly been evaluated in similar clinical

settings within that geographical region.[10] The M4 model was updated in a larger cohort of women with PUL in 2016 with an option to include serum progesterone as a predictor (M6P) or not (M6NP).[13] Recent research suggests that both M6P and M6NP are superior to M4 when compared in EPAUs in the UK.[14] The performance of a prediction model is influenced by the characteristics of the underlying population and may not be generalisable to a different clinical context in another country with the risk of being harmful if implemented.[15] For instance when M4 was tested in a US emergency department the sensitivity for identifying an EP was only 55%.[16] M6NP and M6P have not been validated in an emergency gynaecology unit.

The primary aim of the present study was to externally validate M6NP in an emergency gynaecology unit and compare the performance to its predecessor M4.

## MATERIALS AND METHODS
### Design and setting
External validation was performed on women with PUL attending the emergency gynaecology unit at the Sahlgrenska University Hospital from 1 January 2011 to 31 December 2013. The emergency gynaecology unit serves patients with various gynaecological problems 24 hours a day, 7 days a week either self-referred or referred from other clinics in the greater Gothenburg area.

The initial examination of women with early pregnancy problems was performed by physicians in the early stage of specialist training and a consultant gynaecologist assisted them on request. Women were monitored with serial measurement of hCG levels whenever the physician was uncertain about the pregnancy location. The second blood sample for hCG measurement was routinely scheduled between 2 and 3 days after the first hCG. Interpretation of hCG development was not based on any specific percentage change but approximations of the doubling time in hCG for a viable IUP and the expected hCG decline for a miscarriage.[17] Our follow-up routine for outpatients' after two hCG measurements has been described in a prior study from our unit.[6] Outpatients with deteriorating symptoms were instructed to re-visit the unit. Depending on the severity of the symptoms and the clinical findings, inpatient evaluation and further investigation by a senior physician was considered.

The study hospital was the only referral site for treatment of EP in the area as opposed to other early pregnancy complications. Laparoscopic intervention was the first line of treatment for EP during the study period. Diagnostic laparoscopy was also performed in patients with a prolonged period of follow-up where hCG levels did not decline and there was no evidence of an IUP. In cases where laparoscopy failed to identify an EP uterine evacuation followed to establish a final diagnosis. Methotrexate was used for persistent trophoblast after initial surgery or as treatment of interstitial or caesarean scar pregnancies.

## PARTICIPANTS
Women presenting to the emergency gynaecology unit with problems in early pregnancy, deemed to have a PUL, were screened for inclusion. All women with a PUL and at least two hCG levels were eligible for inclusion into the study. Also, women that were initially managed as outpatients but the second hCG was taken during or after a hospital stay (inpatients) were included unlike prior studies.

In the original M6NP study the second hCG level was imputed when taken more than 3 days after the first hCG.[13] However, in clinical practice the second hCG has only been used for prediction if taken 48±8 hours after the first hCG.[3] We decided to include women with two hCGs taken at least 24 hours and no longer than 72 hours apart, to assure that the validation cohort represented the actual PUL management in our clinic. We excluded women where the final outcome of the PUL was unknown (lost to follow-up).

## DATA COLLECTION
The following variables were obtained from the patients' electronic medical record: Age at the first visit; symptoms related to the first visit (ie, pain or bleeding); prior EP or use of a contraceptive intrauterine device; level of the first and second serum hCG (IU/L); date and time of hCG measurement; type of PUL and the final outcome of the pregnancy. All serum hCG levels were analysed in the same laboratory.

We defined three types of a PUL based on TVUS at the first visit or during follow-up[18]: no signs of a pregnancy neither inside nor outside of the uterus defined a true PUL; when an intrauterine sac-like structure or suspicion of retained products of conception was seen it was defined as a probable IUP; probable EP was used when an ectopic sac-like or solid structure was seen.[19] A gestational sac containing a yolk sac with or without an embryo seen with TVUS defined a definite IUP or EP and therefore not a PUL.[18]

## PREDICTION MODELS
M4 and M6NP use polynomial logistic regression to compute probabilities for each of the three possible outcomes of a PUL: failed PUL, IUP and EP. The covariates of both models are based on two hCG levels analysed in serum (online supplemental table 1).

We used the original equations (online supplemental table 1) of M6NP and M4 to compute probabilities as recommended instead of refitting the model.[15 20] M6P was not validated since serum progesterone is not used in our unit when managing PUL.

## REFERENCE STANDARD
The reference standard was based on the combination of hCG, TVUS findings and surgical procedures. There

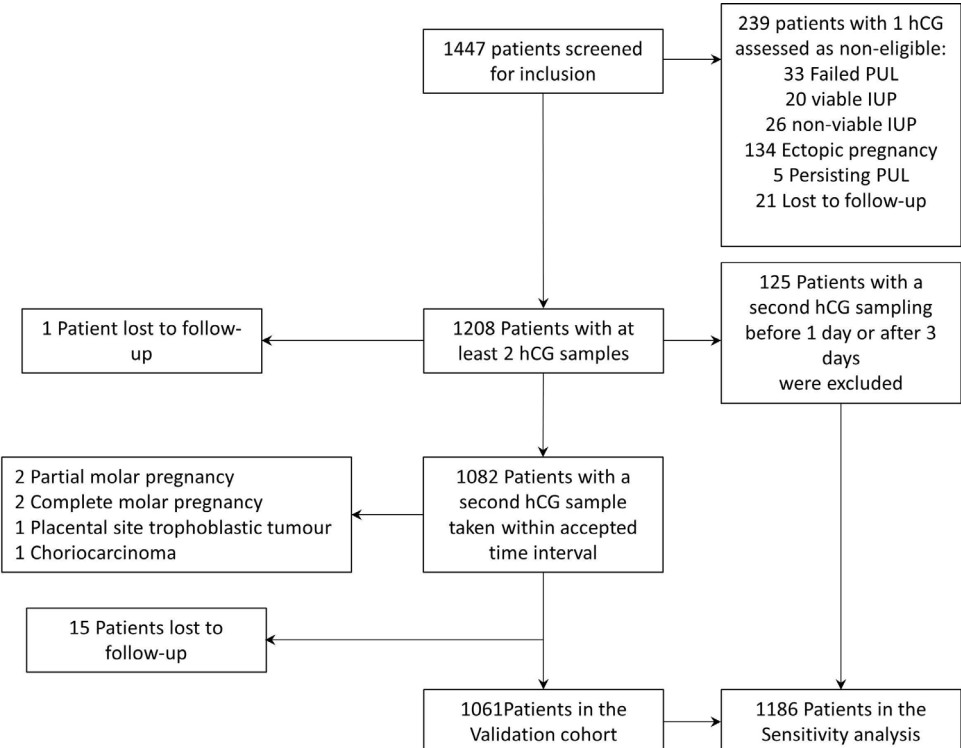

**Figure 1** Flow chart showing the inclusion of patients in the validation cohort and the sensitivity analysis. hCG, human chorionic gonadotropin; IUP, intrauterine pregnancy; PUL, pregnancy of unknown location.

are four categories of PUL outcome and these were set as reference standard: (1) A failed PUL was defined by spontaneously declining hCG levels <0.6 IU/L or a negative pregnancy test during the follow-up where nor an IUP or EP could be confirmed. (2) An IUP was diagnosed by either TVUS; histological findings of chorionic villi after vacuum aspiration; completion of a suspected miscarriage after medical or surgical treatment verified by TVUS or decline of hCG levels <0.6 IU/L. (3) An EP was diagnosed with TVUS, at laparoscopy or hysteroscopy with or without confirmed histology. (4) A persisting PUL (PPUL) was defined when no pregnancy was visible on TVUS and hCG levels plateaued (at least three consecutive hCGs with a change <15% between measurements during a 1-week follow-up). If no chorionic villi were present in uterine aspiration specimen or none was analysed and hCG levels remained elevated also defined a PPUL regardless of prior hCG development as were non-visualised pregnancies treated with methotrexate

In the analysis, PPUL were incorporated among EP as in previous studies.[4 11 13] We used the same definitions as in the original M4 and M6NP studies and a consensus statement.[10 13 18]

The TRIPOD (Transparent Reporting of a multivariable for individual Prognosis or Diagnosis) guidelines were followed conducting this study.[21]

## STUDY SIZE
A validation cohort should contain at least 100 events and ideally 200 events to adequately evaluate prediction rules.[22] Our validation cohort contained 238 EP satisfying this requirement.

## Patient and public involvement statement
Patients and the public were not involved in the development of the research question.

## STATISTICAL ANALYSIS
The predictive performance of a model was evaluated with discrimination and calibration.[23] We focused on the dichotomous outcome of EP and non-EP (IUP and failed PUL) as being the most important to distinguish in clinical practice. Discrimination refers to how well the predicted probabilities of a model differentiates between EP and non-EP and was expressed as the area under the curve (AUC) of the summary receiver operating characteristic (ROC) estimates.

Calibration describes the agreement between the predicted risk of an EP and the observed risk (proportion) of EP.[23] Calibration was assessed by calibration plots, calibration-in-the-large and calibration slope.[24] The calibration plot shows prediction accuracy for each patient individually with a computed loess smoothing curve.[25] Calibration-in-the-large evaluates the average predicted risk of EP in the entire cohort compared with the actual rate of EP while calibration slope evaluates potential underfitting or overfitting of the risk predictions. If the calibration-in-the-large is >0 the average predicted risk of EP is lower than the actual proportion of EP in the cohort, and if the calibration-in-the-large is <0 the

average predicted risk is higher. A calibration slope <1 indicates that (on average) high risk predictions are over-estimated and low risk predictions are underestimated. A slope >1 indicates that (on average) high risk predictions are underestimated and low risk predictions are overestimated.[26]

Classification accuracy was measured with sensitivity (proportion of EP classified as high risk), false positive rate (FPR, proportion of non-EP classified as high risk) and the negative predictive value (NPV, proportion of non-EP among women classified as low risk) as it was presented in the original M6NP study.[13] In implementation studies from EPAU in the UK a PUL has been classified as high risk of EP if the predicted probability of EP reached a threshold (cut-off) of 5%.[14 27] An optimal threshold can vary between clinical settings; we therefore explored other plausible thresholds (2.5%, 5% and 10%) for classification as recommended when validating prediction models.[28]

To determine which model that would lead to the best clinical outcomes (clinical utility) if used for clinical decision-making the net benefit (NB) was measured at the 5% threshold and over a range of thresholds in a decision curve.[29] The NB is calculated as the difference between the number of true positives (correctly identified EP) and false positives (misclassified non-EP) where the latter is weighted by the probability threshold (online supplemental table 1). The choice of a certain threshold for classification reflects how the clinician values a true positive in relation to a false positive. A lower threshold implies that a true positive is valued higher than if a higher threshold is used. For instance, the 5% (odds 1:19) threshold translates to an acceptance of 19 false positives per correctly identified EP. The models' NB was compared with default strategies of assuming all as positive (high risk of EP) and none as positive in a decision curve. If a model has an NB below a default strategy at a specific threshold it has no clinical utility at that threshold.

Since we used a complete case analysis for the validation cohort we performed a sensitivity analysis to test if the exclusion of women with a sampling interval between two hCGs shorter than 24 hours or longer than 72 hours presented a risk for introducing bias.[30] The sensitivity analysis thus included all eligible women regardless of hCG sampling interval.

Performance measures are presented with 95% CIs. McNemar's test was used for testing differences of paired proportions. DeLong's test was used for testing the difference in AUC. A p value of<0.05 was considered statistically significant.

Statistical analyses were performed using SPSS V.21.0 (SPSS, Chicago, Illinois, USA) and SAS V.9.4 (Cary, North Carolina, USA).

## RESULTS

A total of 1447 patients were screened for inclusion into the study. Among them, 239 patients had only one hCG

**Table 1** Descriptive statistics of patients in the validation cohort and patients lost to follow-up

| Characteristics | Validation cohort n=1061 | Lost to follow-up n=15 |
|---|---|---|
| Age (years) | | |
| Median (Q1;Q3) | 31 (26;35) | 33 (29;36) |
| Range | 15–49 | 20–44 |
| First hCG value (IU/L) | | |
| Median (Q1;Q3) | 703 (210;2312) | 640 (272;6325) |
| Range | 7–180 000 | 19–2800 |
| Second hCG value (IU/L) | | |
| Median (Q1;Q3) | 635 (150;2700) | 810 (120;8100) |
| Range | 1–180 000 | 21–3500 |
| hCG ratio (second hCG/first hCG) | | |
| Median (Q1;Q3) | 0.94 (0.45;1.59) | 1.1 (0.79;1.53) |
| Range | 0.06–5.65 | 0.23–1.79 |
| Hours between two hCG samples, n (%) | | |
| <24 | 0 (0) | 0 (0) |
| 24–39 | 141 (13) | 2 (13) |
| 40–56 | 751 (71) | 7 (47) |
| 57–72 | 169 (16) | 3 (20) |
| >72 | 0 (0) | 0 (0) |
| Type of PUL, n (%) | | |
| True PUL | 793 (75) | 11 (73) |
| Probable IUP | 239 (22) | 4 (27) |
| Probable EP | 29 (3) | 0 (0) |
| Vaginal bleeding, n (%) | 723 (68) | 8 (53) |
| Prior ectopic pregnancy, n (%) | 76 (7) | 1 (6) |
| IUD, n (%) | 30 (3) | 0 (0) |

Because of rounding the summarised percentage sometimes are not exactly 100%.
EP, ectopic pregnancy; hCG, human chorionic gonadotropin; IUD, intrauterine device; IUP, intrauterine pregnancy; PUL, pregnancy of unknown location; Q1, first interquartile; Q3, third interquartile.

sample of which 186 had their pregnancy located (134 EP and 46 IUP) before a second hCG was taken (figure 1). Another 33 patients did not have a second hCG since a miscarriage was suspected, although not confirmed, and a urine pregnancy test was used as follow-up. Only 21 patients were lost to follow-up. All 239 patients with a missing second hCG were managed as non-eligible (figure 1). Descriptive statistics of these women are presented in online supplemental table 2.

There were 1208 eligible women. However, 125 (10%) of these had a second hCG outside the targeted time frame. Although excluded from the validation cohort they were part of the sensitivity analysis (figure 1). Six women were diagnosed with a gestational trophoblastic disease and were not part of the validation cohort (figure 1). The validation cohort consisted of 1061 women. Descriptive statistics of the validation cohort and women that were eligible but lost to follow-up (n=15) are presented in table 1. In the validation cohort 71% of the women (751/1061) had an hCG sampling interval of 48±8 hours. The clinical picture warranted inpatient surveillance

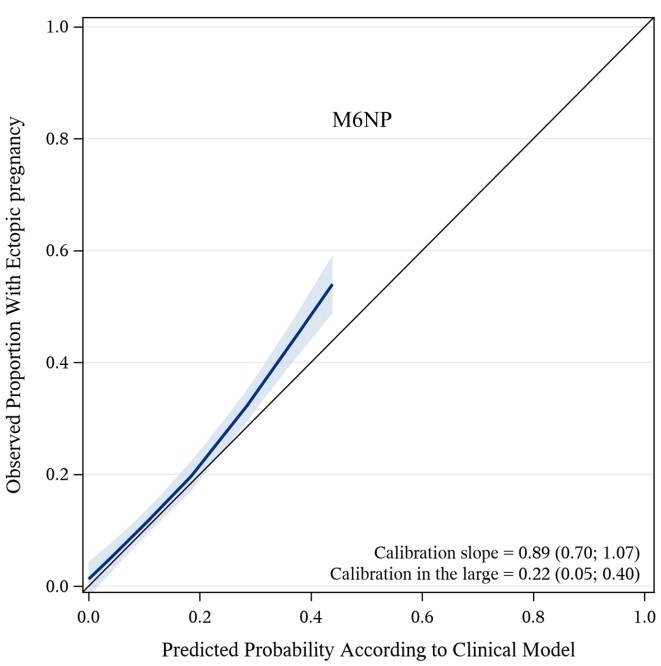 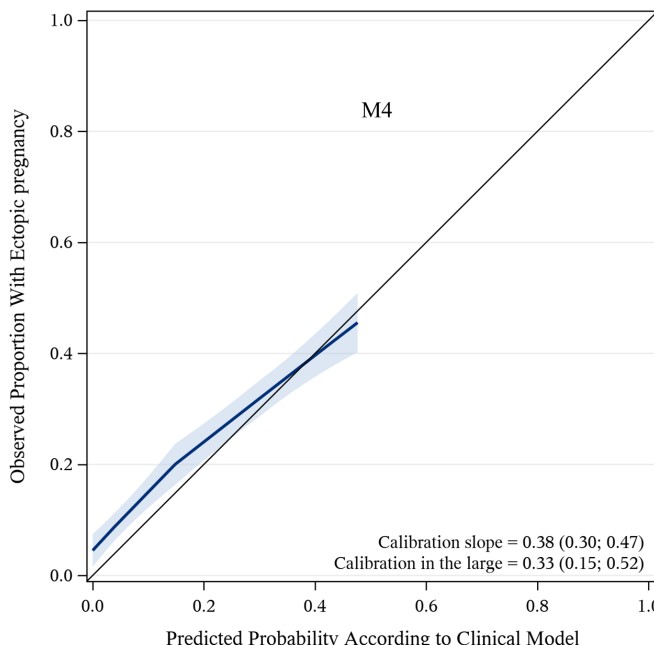

**Figure 2** (A) The calibration plot shows that the curve for M6NP, with pointwise 95% CI (shaded blue area), is close to the line at 45 degrees up to predicted probabilities of 0.20. This indicates that risk predictions for ectopic pregnancies (EPs) are accurate in this range. When the predicted risk is 40% however the observed proportion of ectopic pregnancy is ~50% which means that risk of EP is underestimated. (B) M4 underestimates risk of EP up to estimates of 0.40 as the curve is clearly above the line at 45 degrees in this range.

before a second hCG was taken for 8% of the women (89/1061). Final outcomes of PUL were 461 failed PUL, 362 IUP of which 98 were non-viable and 238 EP (22%) (including 46 PPUL).

## DISCRIMINATION AND CALIBRATION

The AUC for EP was higher for M6NP, 0.85 (95% CI 0.82 to 0.88) than for M4, 0.81 (95% CI 0.78 to 0.84) (p<0.01). As seen in the calibration plot, M6NP was well calibrated up to predictions of 0.20 with the curve close to the diagonal line, but underestimated risk thereafter (figure 2A). M4 made poorly calibrated risk predictions of EP up to 0.40 (figure 2B). Calibration-in-the-large showed that the average predicted risk of EP by both models was underestimated, especially M4 as it was further from 0 as compared with M6NP (figure 2). The average predicted risk by M4 among higher risk groups was overestimated and underestimated among lower risk groups as indicated by the calibration slope being <1, the slope was below but closer to 1 for M6NP (figure 2).

## CLASSIFICATION AND CLINICAL UTILITY

At the 5% risk threshold M6NP had higher sensitivity for identifying an EP, 95% (95% CI 92 to 98) than M4, 85% (95% CI 81 to 90) (p<0.01) as presented in table 2. The FPR was 37% for M4 and 50% for M6NP (p<0.01). The NPV for M4 was 94% and 97% for M6NP (p<0.01). M4 and M6NP classified 52% (95% CI 50 to 55) and 40% (95% CI 37 to 42) of PUL as low risk, respectively (p<0.01). Less than 50% of the failed PUL and 75% of the IUP rendered

risk estimates ≥5% of an EP by M6NP (online supplemental figure 1). This equals the proportion of failed PUL and IUP being misclassified as EP by M6NP. M6NP misclassified 12 EP as low risk of which 7 were predicted to have a failed PUL. Among these EPs, one had emergency surgery due to rupture during follow-up, but no blood transfusion was needed. Online supplemental table 4 presents details for all EP misclassified by M6NP.

At the 10% threshold M6NP had a sensitivity of 91%, FPR of 39% and NPV of 96% (table 2). The sensitivity was lower at 80% for M4 and the FPR was improved but not as marked as for M6NP. At the 2.5% threshold as compared with the 5% threshold the sensitivity and NPV were only marginally higher for M6NP while the FPR increased from 50% to 61% and the proportion of PUL classified as low risk decreased from 40 to 31% (table 2).

Decision curve analysis showed that the curve for M6NP was above that of M4 and the default strategy of assuming all as positive over the entire range of thresholds (figure 3). This means that M6NP had higher NB than M4 and utility at any threshold. M4 had no utility at risk thresholds ≤5% as the curve was under the reference curve of assuming all women as positive.

## SENSITIVITY ANALYSIS

In the sensitivity analysis all women with at least two hCG and any sampling interval were included. Descriptive statistics for all 1186 women were similar with those in the validation cohort except the hCG sampling interval (online supplemental table 5). Both models had marginally lower

**Table 2**  Classification accuracy of the M6NP and M4 at the 2.5%, 5% and 10% threshold

| Threshold | Accuracy measure | M6NP | M4 | P value |
|---|---|---|---|---|
| 2.5% | Sensitivity | 97 (94 to 99) | 91 (88 to 95) | <0.01 |
| | False positive rate | 61 (58 to 65) | 43 (40 to 46) | <0.01 |
| | Negative predictive value | 98 (96 to 99) | 96 (94 to 97) | <0.01 |
| | Percentage classified as low risk | 31 (28 to 34) | 46 (44 to 50) | <0.01 |
| 5% (used in previous studies) | Sensitivity | 95 (92 to 98) | 85 (81 to 90) | <0.01 |
| | False positive rate | 50 (47 to 54) | 37 (34 to 40) | <0.01 |
| | Negative predictive value | 97 (96 to 99) | 94 (92 to 96) | <0.01 |
| | Percentage classified as low risk | 40 (37 to 42) | 52 (50 to 55) | <0.01 |
| 10% | Sensitivity | 91 (87 to 94) | 80 (75 to 85) | <0.01 |
| | False positive rate | 39 (36 to 43) | 30 (37 to 34) | <0.01 |
| | Negative predictive value | 96 (94 to 98) | 92 (90 to 94) | <0.01 |
| | Percentage classified as low risk | 49 (46 to 52) | 58 (56 to 61) | <0.01 |

Reported as percentage (95% CI).

AUC. Calibration for both models was little changed as seen in the calibrations plots (online supplemental figure 2). The sensitivity and the FPR at the 5% risk threshold were almost the same as in the validation cohort for both models, although slightly lower (online supplemental table 6).

## DISCUSSION

This external validation study found that M6NP performs similarly in a Swedish emergency gynaecology unit as in EPAU in the UK where it was developed. M6NP made more accurate predictions of an EP and could better differentiate an EP from a non-EP than M4. A difference between the models in a clinical perspective was that M6NP classified fewer PUL as low risk than M4 leading to its higher sensitivity and FPR at the 5% threshold. This could result in more women being part of some sort of close monitoring if clinical decision-making is based on M6NP as compared with M4, but importantly an opportunity for early diagnosis of more EP starting as PUL. At the

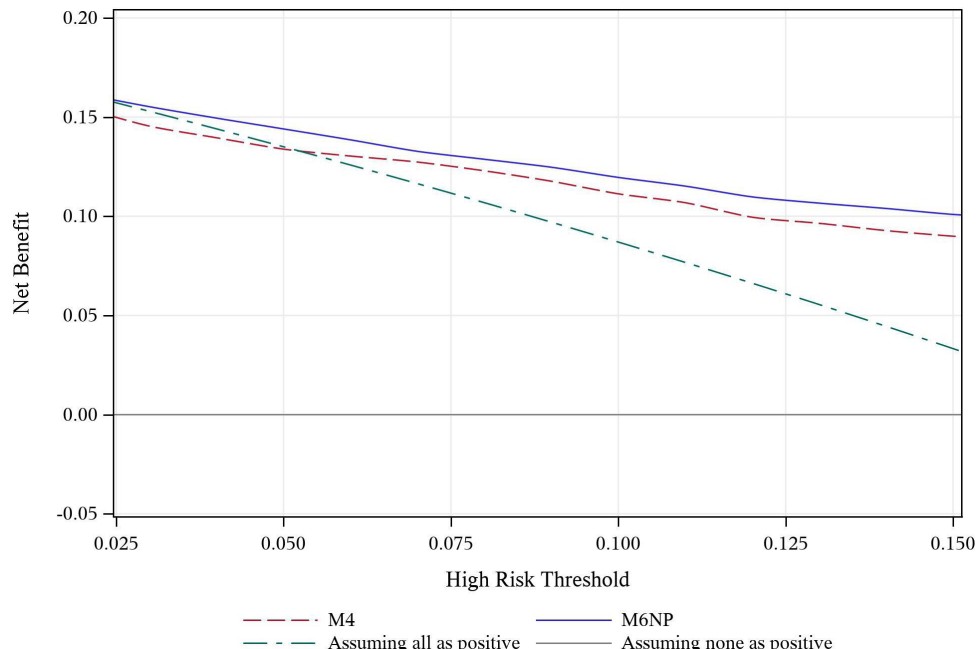

**Figure 3**  Decision curve analysis for M4 (dashed red line) and M6NP (blue line). The high risk threshold is above which a pregnancy of unknown location is classified as high risk (positive) of ectopic pregnancy. M6NP has a higher net benefit compared with M4 and default strategies of assuming all (dashed green line) or none (grey line) of pregnancy of unknown location as high risk over the entire range of thresholds. M6NP thus has utility at any threshold as compared with M4 that has no utility ≤5%.

10% threshold M6NP maintained a high sensitivity while the FPR was significantly lower. M6NP had clinical utility at any threshold unlike M4 that had no clinical utility at thresholds ≤5%.

This is the first study to test the generalisability of M6NP in an emergency gynaecology unit which is an important step before a prediction model can be recommended for, or discouraged from wider implementation.[31] We present real world data on how M6NP would perform in a setting that in several aspects are different from the original cohort with encouraging results.

One limitation of the study was the complete case analysis approach. However, a sensitivity analysis that included all women regardless of sampling interval between two hCGs revealed that the performance of M6NP was sparsely affected. We have no reason to believe that the validation cohort is not representative of the target population and the sample size after exclusion was still large enough for inference about the models' predictive performance.[21] We refrained from including patients lost to follow-up in the sensitivity analysis which could be a limitation of the study. However, given the small fraction of patients lost to follow-up, a sensitivity analysis using multiple imputation would produce virtually identical results as the current analysis.

Six eligible women with a gestational trophoblastic disease were excluded. In a clinical situation a diagnosis is not known at the moment of prediction, why it might have been ideal to include these women. A post hoc analysis revealed that five out of six would have been classified as high risk, affecting specificity negatively since being non-EP. Although a limited number, this must be taken into consideration when interpreting our data. Another limitation was the retrospective design of the study although common when validating or developing prediction models as in the original M6NP study.[15]

The average predicted probability of EP in the entire cohort (calibration-in-the-large) was underestimated, especially for M4, which was an effect of the lower rate of EP in the original cohorts.[10 13] M6NP was well calibrated up to predictions of 0.20 contrary to M4. Instead predictions by M4 were clearly too low up to predictions of 0.40, also reported when recently validated in EPAUs in the UK.[14] This implies that predictions made by M6NP unlike M4 were accurate at thresholds relevant for clinical decision-making and underestimation at higher thresholds may thus be irrelevant in this respect.[23]

M6NP mainly misclassified IUP and to a lesser extent failed PUL as EP in our cohort, consistent with the findings in the UK validation study.[14] M6NP generated higher risk estimates of an EP for failed PUL in our cohort as compared with failed PUL in the validation study from the UK. A greater number of failed PUL was consequently misclassified as high risk at the 5% threshold leading to a higher FPR in our study than in previous studies.[13 14] The reasons for discrepancies between risk predictions could be because of differences in hCG sampling interval, inclusion of inpatients and that the median hCG levels in

our study were higher. The use of imputed hCG values when the second hCG was taken after more than 3 days or missing completely in the UK validation study could also contribute to differences.[14]

We did not assess M6P (adding progesterone as a predictor) since progesterone is not used in our clinic. As previously reported in studies from the UK and Australia it is probable that M6P would perform better as compared with M6NP also in our unit.[14 32]

In clinical practice a woman with a predicted failed PUL after just two hCG measurements is followed up with a home urine pregnancy test after 2 weeks and if an IUP is predicted the examination is repeated after 1 week. Women with a PUL classified as high risk of EP are re-examined within 24 hours.[3 33] In such a clinical situation high classification accuracy is crucial for a prediction model to be useful. In our study M6NP only misclassified 5% (n=12) of the EP as low risk. On the other hand, 50% of non-EP were misclassified as high risk which could result in high numbers of unnecessary visits, admissions to the ward and potentially interventions if implemented. There is insufficient data to rely on when it comes to costs and benefits of misclassifying women with a PUL which makes it difficult to define an optimal threshold for high risk classification, a general challenge when implementing a clinical prediction model.[28] When M4 and M6NP have been used in clinical practice it has been considered reasonable to allow 19 misclassified non-EP for every correctly classified EP which equals the 5% risk threshold.[3 12] Our population had a high rate of EP and the 5% threshold therefore seems justifiable to optimise sensitivity. However, the sensitivity for EP remained high for M6NP at the 10% threshold while the FPR decreased from 50% to 39%. The trade-off between sensitivity and FPR at the 2.5% threshold was less favourable since the FPR increased significantly and the sensitivity was only marginally higher as compared with the 5% threshold. Since M6NP had clinical utility at the 5 and 10% thresholds we suggest that any of these are appropriate to use in our clinical setting.

Regardless of which threshold is finally implemented one must remember that M6NP should always be used as a complement to clinical judgement and never as a sole decision instrument. M6NP and M6P are available at www.earlypregnancycare.co.uk where the 5% threshold is used by default.

## Conclusion

The performance of M6NP as opposed to M4 was encouraging when validated on women with PUL in an emergency gynaecology unit. M6NP can be introduced in related yet untested clinical settings outside the UK and EPAU to effectively support clinical decision-making. Updating M6NP in a local context to account for geographical differences could be an option to improve calibration.

**Contributors** All persons listed as authors fulfil the International Committee of Medical Journal Editors (ICMJE) criteria for authorship: JF contributed substantially

to conception and design of the study, interpreted the data and drafted the article and approved the final version after revision. JF is responsible for the overall content as guarantor. CB commented on the drafts, made revisions to the article and approved the final version. AS contributed substantially to conception and design of the study, interpreted the data, wrote the article and approved the final version. All authors agree to be accountable for all aspects of the work in ensuring that questions related to the accuracy or integrity of any part of the work are appropriately investigated and resolved.

**Funding** This work was supported by grants from the Swedish state under the agreement between the Swedish government and the county councils, the ALF-agreement (ALFGBG-720291, ALFGBG-72150) and The Healthcare Board, Region Västra Götaland (VGFOUREG-660661).

**Competing interests** None declared.

**Patient and public involvement** Patients and/or the public were not involved in the design, or conduct, or reporting, or dissemination plans of this research.

**Patient consent for publication** Not applicable.

**Ethics approval** This study involves human participants and was approved by The Regional Ethical Review Board, Gothenburg, Sweden (approval # 50115). This was an observational study with retrospective data and did not require patient consent when approved by the regional ethical review board.

**Provenance and peer review** Not commissioned; externally peer reviewed.

**Data availability statement** Data are available upon reasonable request. Data are available on reasonable request. Requests should be made to the lead author johan.fistouris@vgregion.se.

**ORCID iD**
Johan Fistouris http://orcid.org/0000-0002-6816-5691

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
