## [Reviewer comments · BMJ Open]

ARTICLE DETAILS

TITLE (PROVISIONAL)	Pregnancy of unknown location: external validation of the hCG based M6NP and M4 prediction models in an emergency gynaecology unit.
AUTHORS	Fistouris, Johan; Bergh, Christina; Strandell, Annika

VERSION 1 – REVIEW

REVIEWER	Van Calster, Ben KU Leuven, Department of Development and Regeneration
REVIEW RETURNED	20-Dec-2021

GENERAL COMMENTS	I reviewed this paper as a medical statistician (and as the statistician who developed the M4 and M6 models). External validation studies of prediction models are highly relevant and still too uncommon; therefore, this effort is clearly of interest. The methodology and reporting of this study was very good, in my opinion. I would like to make some suggestions that may help the authors to improve the manuscript. Key comments 1. There is an M6 version with progesterone and a version without progesterone. The authors should be clear regarding the version they used.2. The study flow starts at 1202 patients with at least 2 hCG measurements. The authors do not mention how many PUL there were without 2 measurements. This should be clarified. Would imputation not be an option, or were there hardly cases with missing second hCG?3. It might be interesting to provide some descriptive statistics for cases that were lost to follow up.4. The description of calibration could be improved in the abstract: M4 seems to consistently underestimate the risk of EP. M6 has well calibrate risk estimates up to estimates of 20%, thereafter there appears to be underestimation as well. (See other comment about the description in the main text.)5. It would be useful to include pointwise confidence intervals for the calibration curves.6. Line 183-185: the description of the calibration slope is not correct. A slope < 1 indicates that (on average) high estimated risks are overestimated and low estimated risks are underestimated. A slope > 1 indicates that (on average) high estimated risks are underestimated and low estimated risks are overestimated.
--

	Minor thoughts and comments  1. Line 130: is there not a fourth type, i.e. probable miscarriage? 2. I am not a clinician, but I was confused by the definition of IUP as an outcome. It seems that seeing a gestational sac with embryo in the uterus is not a necessary condition? 3. Line 157: 5% is not the risk threshold that belongs to the model. A risk model is independent from a risk threshold that you define for it. Please reformulate. 4. Line 202-204: the NB for the model should not only be above the NB for classifying everyone as at high risk, but should also be positive. (If NB is negative at a given threshold, the utility of the model with that given threshold is lower than the utility of classifying everyone as at low risk.) 5. McNemar's test: this for comparing paired 'proportions' (not just paired 'samples') 6. Software: first introduce the software packages used, then mention the val.prob.ci.2 function. Also, make clear that val.prob.ci.2 and rmda are for R. 7. Line 225-234: could this not be summarized in a more concise manner? E.g. the hCG levels in your sample tended to be slightly higher than the hCG levels in the M6 development paper. 8. Line 238: was the p-value for the difference in c-statistic based on a McNemar test? That would not be appropriate. Or was this based on another method (most often the DeLong method is used). You could also use your bootstrapping procedure to give a 95% confidence interval for the difference in c-statistic. 9. Line 238-244: For an external validation study with decent sample size (as in this case), the calibration curve is the key result, and is more important than the calibration intercept/slope. Therefore, the two last sentences (line 241-244) are the key finding here. In addition, the interpretation of the calibration slope is not correct (cf other comment on the calibration slope). 10. Line 258: 'M4 had no NB' should be 'M4 had no utility'. 11. Underestimation of estimated risks is not relevant for decision making: this seems correct. However, is patient counseling an issue here (again, I am not a clinician so I cannot judge this)? If so, some model updating could be tried. 12. Line 281-283: Christodoulou et al imputed cases with missing second hCG and those who were lost to follow-up. This could also contribute to differences, I think. 13. Line 283-284: I do not understand what the authors mean with "combining these differences ultimately led to a stronger test of the generalizability of M6". 14. Line 304-305: it is unclear what the authors mean with the theory of NB and DCA. The optimal threshold cannot be based on NB or DCA 'results', I would avoid to give this impression. 15. Table 1: why are there higher frequencies for intervals of 1 day and 3 days in the validation cohort vs all patients? 16. Model formulas: clarify that log refers to the natural logarithm.
--	---

REVIEWER	Ludlow, Joanne Royal Prince Alfred Hospital, Women and Babies
REVIEW RETURNED	23-Dec-2021

GENERAL COMMENTS	The subject matter is of great interest to those practising in early pregnancy units (EPU) and emergency gynaecology settings. External validation of the published mathematical models in the management of Pregnancy of unknown location (PUL) in an
--

emergency gynaecology unit is a commendable aim. However, there are several limitations in this study/manuscript.

There is no documentation of ethics approval being granted for this study.

The index period for this study is from 2011 to 2013, eight to 10 years ago. This is outdated and the M6(P) mathematical model (including serum progesterone) validation for classification of PUL has been published in the intervening period.

The references are very comprehensive. But there is a more recent update than Reference 1. This is <https://www.nice.org.uk/guidance/ng126>

The abstract does not include a methods section. The manuscript is too long, particularly the statistics section.

Pregnancy of unknown location has not been adequately defined.

The PUL rate based on the initial ultrasound scan is not reported in this paper.

Were all the women who had PUL in the index period included in the study? If not, which women were excluded and what were the reasons for their exclusion? What was the total number of women with early pregnancy complications who presented during this time? A flow chart of women who presented with all early pregnancy complications such as known intrauterine and extrauterine pregnancies and PUL. All PUL and their subsequent outcomes, any exclusions (with explanatory reasons) should be included. The Flow Diagram on page 23 does not include enough of this detail.

The serum BHCG interval is reported in days rather than hours. Did all the women have a serum BHCG interval of 48 hours? What were the minimum and maximum serum BHCG intervals?

Were all the serum BHCG levels taken in the same laboratory?

Is progesterone used in your unit as part of your assessment of PUL?

Does your unit have a defined protocol for the management of PUL? And is this always followed? Were there any protocol violations? And if so, were they based on the clinical situation, such as concerns about an ectopic pregnancy and the need for immediate surgery? Were there any ruptured ectopic pregnancies in the women classified in the low-risk group

The statistics in this manuscript would require a formal statistical review. Why was the “c-statistic” rather than area under the curve (AUC) for the receiver operator curves (ROC)?

What do you mean by one of the limitations was the “complete case analysis with a risk for introducing selection bias; however, a sensitivity analysis showed consistent predictive performance and classification accuracy”? Do you mean that the women who only had one serum BHCG were still correctly classified regarding their risk of ectopic pregnancy? Why were they included at all when you were doing an external validation of the originally described M4 and M6 mathematical models from the London and Belgium Early Pregnancy Units (EPU) which did not include those women who only had one serum BHCG?

The description of TRIPOD and some of the statistical analysis is overly complicated. The optimal threshold (5% for high risk of ectopic pregnancy or persistent pregnancy of unknown location (PPUL)) may vary in each population, but you do not explain why this is relevant in your population. A clearer explanation of “net benefit” is required.

The prevalence of ectopic pregnancies in this cohort is higher than generally seen in EPU and as such the positive predictive value of

	the models would be expected to be higher than in the original cohort. On page 8, line 134 a “corneal” pregnancy is written. Clearly, you mean cornual. In your unit is interstitial and cornual used interchangeably or do you regard them as being in different locations? ESHRE (European Society for Human Reproduction and Embryology) have an expert consensus agreement about these definitions. (Reference - Human Reproduction Open, Vol.00, No.0, pp. 1–21, 2020 doi:10.1093/hropen/hoaa055). This is attached Table 1 is complicated and the data for age and serum BHCG parameters is extremely unclear, containing too many, unexplained figures. Table 2 is very good and very clear The Figures on pages 27 and 28 do not add to the manuscript and should be removed. This paper is not publishable in its current form. Rationalisation of some of the statistics, more editing of the manuscript, written confirmation of ethics approval, more upto-date data and references, fewer figures, tables and graphs, a clearer table 1, including the M6(P) protocol would make the manuscript more likely to be accepted for publication.
--	---

VERSION 1 – AUTHOR RESPONSE

Reviewer: 1

1. There is an M6 version with progesterone and a version without progesterone. The authors should be clear regarding the version they used.

Response: We have clarified which version of M6 that we used in our study and why M6P was not part of our study and we now refer to M6NP.

The following four texts have been added to the manuscript:

The primary aim of the present study was to externally validate M6NP in an emergency gynaecology unit and compare the performance to its predecessor M4. (lines 117-118)

The M4 model was updated in a larger cohort of PUL in 2016 with an option to include progesterone as a predictor (M6P) or not (M6NP). (lines 101-102)

M6P was not validated since progesterone was not used in our unit when managing PUL. (lines 187-188)

We did not assess M6P (adding progesterone as a predictor) since progesterone is not used in our clinic. As demonstrated in recent studies from EPAU in both the UK and Australia it is likely that M6P would perform better as compared to M6NP also in an emergency gynaecology unit [14, 32]. (lines 402-405)

2. The study flow starts at 1202 patients with at least 2 hCG measurements. The authors do not mention how many PUL there were without 2 measurements. This should be clarified. Would imputation not be an option, or were there hardly cases with missing second hCG?

Response: We have expanded the flowchart to include those with only one hCG. Considering the large original sample size, we believe that imputation might not be of great importance. Furthermore, among patients with only one hCG, the majority was admitted directly to in-hospital care and received immediate or fast management due to the clinical course and were thus not eligible for a second hCG. Since these hCG might not have been missing at random, imputation could possibly instead introduce bias. Thus, we would prefer to refrain from imputation.

The following has been added in ms: *We excluded women where only one hCG was taken or the final outcome of PUL was unknown (lost to follow-up). (lines 160-161)*

3. It might be interesting to provide some descriptive statistics for cases that were lost to follow up.

Response: We have provided data on women that were lost to follow up in Table 1.

4. The description of calibration could be improved in the abstract: M4 seems to consistently underestimate the risk of EP. M6 has well calibrated risk estimates up to estimates of 20%, thereafter there appears to be underestimation as well. (See other comment about the description in the main text.)

Response: Thank you for improving the description. We have added the following text: M6NP made accurate risk predictions of ectopic pregnancies up to estimates of 0.20 but underestimated risk thereafter. M4 underestimated risk up to estimates of approximately 0.40. (lines 43-46)

5. It would be useful to include pointwise confidence intervals for the calibration curves.

Response: Our Statistician used SAS to compute a new calibration plot with pointwise 95% confidence intervals (Fig. 2). We have revised the Statistics section, please see point 6., below.

6. Line 183-185: the description of the calibration slope is not correct. A slope < 1 indicates that (on average) high estimated risks are overestimated and low estimated risks are underestimated. A slope > 1 indicates that (on average) high estimated risks are underestimated and low estimated risks are overestimated.

Response: Thank you for bringing this to our attention. We have made changes to the description of the slope and added the following text: A slope < 1 indicates that (on average) high risk predictions are overestimated and low risk predictions are underestimated. A slope > 1 indicates that (on average) high risk predictions are underestimated and low risk predictions are overestimated. (lines 234-237)

Minor thoughts and comments

1. Line 130: is there not a fourth type, i.e. probable miscarriage?

Response: Probable miscarriages are included among probable IUP, although not discernible. We used three types/categories of PUL that were described in a consensus statement (Barnhart, 2011).

2. I am not a clinician, but I was confused by the definition of IUP as an outcome. It seems that seeing a gestational sac with embryo in the uterus is not a necessary condition?

Response: The presence of a yolk sac is enough for diagnosing an IUP with ultrasound, also described in the consensus paper; we added the reference (line 168). However, some IUP are not diagnosed with ultrasound but following vacuum aspiration when chorionic villi are present in the pathological analysis and a part of the reference standard.

3. Line 157: 5% is not the risk threshold that belongs to the model. A risk model is independent from a risk threshold that you define for it. Please reformulate.

Response: We removed the description about the risk threshold. However, we added the following text in the statistics section:

In implementation studies from EPAU in the UK a PUL has been classified as high risk of EP if the predicted probability of EP reached a threshold of 5% [14, 27]. An optimal threshold can vary between clinical settings; we therefore explored other plausible thresholds (2.5, 5, and 10%) for classification as recommended when validating prediction models [28]. (lines 243-245)

4. Line 202-204: the NB for the model should not only be above the NB for classifying everyone as at high risk, but should also be positive. (If NB is negative at a given threshold, the utility of the model with that given threshold is lower than the utility of classifying everyone as at low risk.)

Response: We have added the explanation of the default strategy of treating everyone as low risk.

Text: In the decision curve a model is compared with default strategies of assuming all as positive (high risk of EP) and none as positive. If a model has a NB below a default strategy at a specific threshold it has no clinical utility at that threshold. (lines 255-257)

5. McNemar's test: this for comparing paired 'proportions' (not just paired 'samples')

Response: We have changed the description.

Text: McNemar's test was used for testing differences of paired proportions. (line 277)

6. Software: first introduce the software packages used, then mention the val.prob.ci.2 function. Also, make clear that val.prob.ci.2 and rmda are for R.

Response: The Statistician consulted for revision of the analyses did not use R and the val.prob.ci.2 function, but instead he used SAS for the construction of the calibration plot and the decision curve. The statistics section has been revised accordingly.

Text: Statistical analyses were performed using SPSS 21.0 (SPSS, Chicago, IL, USA) and SAS version 9.4 (Cary, NC, USA). (lines 280-281)

7. Line 225-234: could this not be summarized in a more concise manner? E.g. the hCG levels in your sample tended to be slightly higher than the hCG levels in the M6 development paper.

Response: We have summarised according to your suggestion.

Text: As seen in supplemental Table 2, the median of the first hCG value was numerically higher and the median hCG ratio was lower for EP in our validation cohort as compared with EP in the original M6NP cohort. Also, for IUP the median of the first hCG was numerically higher but the median hCG ratio was lower in our cohort. Differences in hCG characteristics appears to be lower for failed PUL when comparing our cohort with the original M6NP cohort. (lines x-x)

8. Line 238: was the p-value for the difference in c-statistic based on a McNemar test? That would not be appropriate. Or was this based on another method (most often the DeLong method is used). You could also use your bootstrapping procedure to give a 95% confidence interval for the difference in c-statistic.

Response: We used the DeLong to test the difference in AUC, this has been clarified. Text: DeLong's test was used for testing the difference in AUC. (line 278)

9. Line 238-244: For an external validation study with decent sample size (as in this case), the calibration curve is the key result, and is more important than the calibration intercept/slope. Therefore, the two last sentences (line 241-244) are the key finding here. In addition, the interpretation of the calibration slope is not correct (cf other comment on the calibration slope).

Response: We now start with describing the calibration plot and then calibration in the large and the slope. The description of the slope has been revised, as mentioned above ("first" point 6).

Text: As seen in the calibration plot M6NP was well calibrated up to predictions of 0.20 with the curve close to the diagonal line, but underestimated risk thereafter (Fig. 2A). M4 made poorly calibrated risk predictions of EP up to 0.40 (Fig. 2B). Calibration in the large showed that the average predicted risk of EP by both models was underestimated, especially M4 as the intercept was further from 0 as compared with M6NP (Fig. 2). The average predicted risk by M4 among higher risk groups was overestimated and underestimated among lower risk groups as indicated by the slope being < 1, the slope was below but closer to 1 for M6NP (Fig. 2). (Lines 315-328)

10. Line 258: 'M4 had no NB' should be 'M4 had no utility'.

Response: We have reformulated as suggested. (line 351)

11. Underestimation of estimated risks is not relevant for decision making: this seems correct.

However, is patient counseling an issue here (again, I am not a clinician so I cannot judge this)? If so, some model updating could be tried.

Response: We agree, updating is absolutely an option to improve calibration which has been mentioned in the conclusion, see conclusion. (line 435)

12. Line 281-283: Christodoulou et al imputed cases with missing second hCG and those who were lost to follow-up. This could also contribute to differences, I think.

Response: We agree, this has been added in the discussion section. Text: Also, the use of imputation when the second hCG was missing completely or when the second hCG was taken more than 3 days after the first hCG could have contributed. (lines 399-401).

13. Line 283-284: I do not understand what the authors mean with "combining these differences ultimately led to a stronger test of the generalizability of M6".

Response: We decided to remove the phrase.

14. Line 304-305: it is unclear what the authors mean with the theory of NB and DCA. The optimal threshold cannot be based on NB or DCA 'results', I would avoid to give this impression.

Response: We have removed the phrase.

15. Table 1: why are there higher frequencies for intervals of 1 day and 3 days in the validation cohort vs all patients?

Response: Typing error, this has been changed.

16. Model formulas: clarify that log refers to the natural logarithm.

Response: This has been clarified in supplemental Table 1.

Reviewer: 2

Dr. Joanne Ludlow, Royal Prince Alfred Hospital

Comments to the Author:

The subject matter is of great interest to those practising in early pregnancy units (EPU) and emergency gynaecology settings. External validation of the published mathematical models in the management of Pregnancy of unknown location (PUL) in an emergency gynaecology unit is a commendable aim. However, there are several limitations in this study/manuscript.

1. There is no documentation of ethics approval being granted for this study.

Response: Ethical approval is already in the manuscript. (lines 445-446)

2. The index period for this study is from 2011 to 2013, eight to 10 years ago. This is outdated and the M6(P) mathematical model (including serum progesterone) validation for classification of PUL has been published in the intervening period.

Response: Since the models' predictions are solely based on two hCG values their performance should not be significantly influenced by a certain time period. As for previous temporal validation of M4 we expect to find consistent results for M6NP in our setting. Also, the development of M6NP was on data from July 2003 to February 2007 and from April 2009 to December 2013. This is before and during our study period. We therefore believe the time period for validating the models are secondary,

3. The references are very comprehensive. But there is a more recent update than Reference 1. This is <https://www.nice.org.uk/guidance/ng126>

Response: The most recent NICE update, as suggested, could be found as Reference 31 in the original manuscript (now Reference 32). We believe that the RCOG reference (reference 1) is also valuable to keep.

4. The abstract does not include a methods section. The manuscript is too long, particularly the statistics section.

Response: We followed the BMJ Open guidance for abstract and manuscript presentation which also included a limitation of the word count. However we have added a methods section in the abstract.

Text: All consecutive women with a pregnancy of unknown location during a study period of 3 years were reviewed for inclusion in the validation cohort. Predictions made by M6NP and M4 were based on two serum human chorionic levels taken with an interval of at least 24 hours and no longer than 72 hours. (lines 27-30)

The statistics section has been shortened.

5. Pregnancy of unknown location has not been adequately defined. The PUL rate based on the initial ultrasound scan is not reported in this paper.

Response: We have defined PUL according to a consensus paper (reference 18). Both probable EP and IUP are managed as PUL in our clinic as in a validation study from the UK that we have used as reference (reference 13-). Women with PUL are registered in a certain file and easily available unlike other pregnancy complications such as miscarriages or normal IUP. To obtain a PUL rate was thus not possible since the denominator was unknown.

6. Were all the women who had PUL in the index period included in the study? If not, which women were excluded and what were the reasons for their exclusion? What was the total number of women with early pregnancy complications who presented during this time? A flow chart of women who presented with all early pregnancy complications such as known intrauterine and extrauterine pregnancies and PUL. All PUL and their subsequent outcomes, any exclusions (with explanatory reasons) should be included. The Flow Diagram on page 23 does not include enough of this detail.

Response: All consecutive women with a PUL were reviewed for inclusion into the validation cohort. There were a number of PUL that had only 1 hCG measurement which has been added to the updated flowchart as well as their final outcomes. We also added women that we excluded from the validation cohort due to a final outcome of a gestational trophoblastic disease.

We also added the following text: A total of 1447 consecutive women with PUL were reviewed for inclusion into the validation cohort as shown in the flowchart (Fig. 1). There were 239 women that only had one hCG taken and six women with a final outcome of a gestational trophoblastic disease and were therefore not included in the study. Another 125 women were excluded due to having an hCG sampling interval outside 24 to 72 hours. Among all 1447 women, 37 were lost to follow-up (lines 286-291)

Women with other early pregnancy complications were unknown, see above point 5. The flowchart only includes women with PUL since the number of other early pregnancy complications was unknown. We have included all women with PUL and their subsequent outcomes unless lost to follow-up. The flow chart has been updated with available details.

7. The serum BHCG interval is reported in days rather than hours. Did all the women have a serum BHCG interval of 48 hours? What were the minimum and maximum serum BHCG intervals?

Response: We categorised 48 ± 8 hours as 2 days, this has been clarified, and we have changed the intervals from days to hours in Table 1. We included women with an hCG sampling interval between 24 and 72 hours.

We added the following text: In the original M6NP study the second hCG level was imputed if it was taken more than 3 days after the first hCG [13]. However in clinical practice the second hCG has only been used for prediction if taken 48 ± 8 hours after the first [3]. We decided to include women with a sampling interval between 24 and 72 hours so that the validation cohort represented the actual PUL management in our clinical setting. We excluded women where only one hCG was taken or the final outcome of PUL was unknown (lost to follow-up). (lines 156-161)

8. Were all the serum BHCG levels taken in the same laboratory?

Response: All serum –hCG levels were analysed in the same laboratory, this has been added in the methods section.

Text: All serum hCG levels were analysed in the same laboratory. (line 167)

9. Is progesterone used in your unit as part of your assessment of PUL?

Response: We do not use progesterone in our clinic, this has been clarified in the methods section and the discussion.

Text: M6P was not validated since progesterone was not used in our unit when managing PUL. (line 186).

Text: We did not assess M6P (adding progesterone as a predictor) since progesterone is not used in our clinic. As reported in recent studies from EPAU in both the UK and Australia it is likely that M6P would perform better as compared with M6NP also in an emergency gynaecology unit [14, 32]. (lines 402-405)

10. Does your unit have a defined protocol for the management of PUL? And is this always followed? Were there any protocol violations? And if so, were they based on the clinical situation, such as concerns about an ectopic pregnancy and the need for immediate surgery? Were there any ruptured ectopic pregnancies in the women classified in the low-risk group.

Response: The aim of our study was not to evaluate our management protocol but it has been described in detail in a previous article (reference 6) and clarified in the text.

Text: Our routine for outpatients' surveillance has been described in a prior study from our emergency gynaecology unit. (lines 134-135)

We have added a supplemental table (Table 3) about the details of the ectopic pregnancies that were misclassified as low risk by M6NP. Although we were reluctant to add more tables considering your opinion about too many tables and figures, yet this particular table will only be available online.

11. The statistics in this manuscript would require a formal statistical review. Why was the "c-statistic" rather than area under the curve (AUC) for the receiver operator curves (ROC)?

Response: We are grateful for having had a formal statistical review from Reviewer 1 and our own Statistician. We have replaced the c-statistic with AUC, which is probably more easily recognised by the readers, although any can be used to express the discriminatory ability of a logistic regression model.

12. What do you mean by one of the limitations was the “complete case analysis with a risk for introducing selection bias; however, a sensitivity analysis showed consistent predictive performance and classification accuracy”? Do you mean that the women who only had one serum BHCG were still correctly classified regarding their risk of ectopic pregnancy? Why were they included at all when you were doing an external validation of the originally described M4 and M6 mathematical models from the London and Belgium Early Pregnancy Units (EPU) which did not include those women who only had one serum BHCG?

Response: We only included women with 2 hCG. However, in the sensitivity analysis we included all women regardless of sampling interval, not only those with an interval between 1 (24h) and 3 days (72h). This has been clarified. With a sensitivity analysis we want to test the robustness of the model performance when the hCG interval varied and if the complete case analysis would bias our results. The following text has been added: In the original M6NP study the second hCG level was imputed if it was taken more than 3 days after the first hCG [13]. However in clinical practice the second hCG has only been used for prediction if taken 48 ± 8 hours after the first [3]. We decided to include women with a sampling interval between 24 and 72 hours so that the validation cohort represented the actual PUL management in our clinical setting.

We excluded women where only one hCG was measured or the final outcome of PUL was unknown (lost to follow-up). (lines 156-161)

13. The description of TRIPOD and some of the statistical analysis is overly complicated. The optimal threshold (5% for high risk of ectopic pregnancy or persistent pregnancy of unknown location (PPUL)) may vary in each population, but you do not explain why this is relevant in your population. A clearer explanation of “net benefit” is required.

Response: At a lower threshold sensitivity increases and specificity decreases. By evaluating multiple thresholds we ought to perhaps find a more preferable combination of sensitivity and specificity than the one given at the 5% threshold.

In the method section this has been clarified with the following text: In implementation studies from EPAU in the UK a PUL has been classified as high risk of EP if the predicted risk of EP reached a threshold of 5% [14, 27]. An optimal threshold can vary between clinical settings; we therefore explored other plausible thresholds (2.5, 5, and 10%) for classification as recommended when validating prediction models [28]. (lines 241-245)

We have elaborated on this in the discussion.

Text: Our population had a high rate of EP and the 5% threshold therefore seems justifiable to optimize sensitivity. However, for M6NP sensitivity remained high at the 10% threshold while the FPR decreased from 50 to 39%. The trade-off between sensitivity and FPR at the 2.5% threshold was less favourable since the false positive rate increased significantly but the sensitivity was only marginally higher as compared with the 5% threshold. M6NP had clinical utility at the 5 and 10% thresholds. We therefore suggest that any of these are appropriate to use in our clinical setting. (lines 421-426)

We have revised the explanation of Net Benefit.

Text: To determine which model that would lead to the best clinical outcomes (clinical utility) if used for clinical decision making the Net Benefit (NB) was measured at the 5% threshold and over a range of thresholds in a decision curve. The NB is calculated as the difference between the number of true positives (correctly identified EP) and false positives (misclassified non-EP) where the latter is weighed by the probability threshold (Supplemental). The choice of a certain threshold for classification reflects how the clinician values a true positive in relation to a false positive. A lower threshold implies that a true positive is valued higher than if a higher threshold is used. For instance, the 5% (odds 1:19) threshold translates to an acceptance of 19 false positives per correctly identified EP. The models' NB was compared with default strategies of assuming all as positive (high risk of EP) and none as positive in a decision curve. If a model has a NB below a default strategy at a specific threshold it has no utility at that threshold. (lines 246-257).

14. The prevalence of ectopic pregnancies in this cohort is higher than generally seen in EPU and as such the positive predictive value of the models would be expected to be higher than in the original cohort.

Response: That could be the case, however a higher false positive rate would contribute to a lower positive predictive value. Positive predictive value was not reported in the original study.

On page 8, line 134 a “corneal” pregnancy is written. Clearly, you mean cornual. In your unit is interstitial and cornual used interchangeably or do you regard them as being in different locations? ESHRE (European Society for Human Reproduction and Embryology) have an expert consensus agreement about these definitions. (Reference - Human Reproduction Open, Vol.00, No.0, pp. 1–21, 2020 doi:10.1093/hropen/hoaa055). This is attached

Response: These are separate entities and are not used interchangeably. We have used your reference to highlight how any ectopic would be defined with ultrasound if/when described in the manuscript.

15. Table 1 is complicated and the data for age and serum BHCG parameters is extremely unclear, containing too many, unexplained figures.

Response: Table 1 has been revised.

Table 2 is very good and very clear

16. The Figures on pages 27 and 28 do not add to the manuscript and should be removed.

Response: We have removed those figures and only kept the one that was referred to in the text.

This paper is not publishable in its current form.

17. Rationalisation of some of the statistics, more editing of the manuscript, written confirmation of ethics approval, more up to-date data and references, fewer figures, tables and graphs, a clearer table 1, including the M6(P) protocol would make the manuscript more likely to be accepted for publication.

Response: We have restricted the number of tables and figures to the journal guideline. In the discussion we have included a couple of new references. Revised Table 1 and had a formal statistical review. Supplemental figures and tables will only be available online for those particularly interested. M6P is not possible for us to validate since progesterone is not used for PUL management in our clinic.

Please also see attachment

Reviewer: 1

Competing interests of Reviewer: I am the statistician who developed the M4 and M6 models.

Reviewer: 2

Competing interests of Reviewer: None

VERSION 2 – REVIEW

REVIEWER	Van Calster, Ben KU Leuven, Department of Development and Regeneration
REVIEW RETURNED	09-Apr-2022

GENERAL COMMENTS	Thank you for the improvements. I still have some minor comments. 1. Please update the URL to www.earlypregnancycares.co.uk, the one currently mentioned does not work.2. Calibration intercept: I am not sure whether this is described or performed correctly. The text refers to ‘the intercept of the curve’, where the curve in the paper is based on loess. However, the calibration intercept refers to the intercept of the logistic model
---

	logit($P(Y=1) = a + LP$, i.e. a model where the slope has been fixed to 1 by including it as an offset. Is this how it is done? 3. About the patients with only 1 hCG level: the authors believe that imputation may lead to bias because the missing data may not be missing at random (MAR). What are the arguments for this? The authors reply that ‘the majority’ of these cases were admitted due to ‘the clinical course’ and were therefore not eligible for a second hCG. What does that mean exactly? Related to this, more than half of these cases were EP/PPUL. So I wonder about the reason for not having a second hCG. E.g., perhaps many of these cases had their pregnancy confirmed on a follow-up ultrasound scan before taking the second hCG? I mean, if the pregnancy is confirmed on day 2, then it is not a PUL anymore and the model has no use. It would be good if the authors could provide details on the reason for not having a second hCG in the paper, and to provide descriptive statistics for these patients. 4. Because of the previous issue, I think the description on line 52 and on line 314 is not complete. The authors mention complete case analysis as a limitation, but the sensitivity analysis only refers to patients where the second hCG is not in the targeted interval, not to patients with missing second hCG or that are lost to follow up. I would like these issues to be discussed as well. 5. Line 248 states that 89 were managed as inpatients. Why was that? 6. Line 270: there is no p-value for sensitivity, but there is one for FPR. Why? 7. Line 273: “rendered 5% risk predictions” means “rendered risk estimates $\geq 5\%$”? 8. The cases with molar pregnancy should ideally not have been excluded. At the moment of making a prediction, it is not known that they are molar. I leave this up to the authors (we have been ‘guilty’ of this before until a reviewer pointed this out...). 9. Line 329: “M6NP mainly misclassified IUP and to a lesser extent failed PUL in our cohort as reported for M6P”, I do not understand this sentence. You cannot simply compare M6NP with M6P?
--	--

VERSION 2 – AUTHOR RESPONSE

Dear Prof. Van Calster,

Thank you for giving us the opportunity to submit a revised draft of our manuscript titled “Pregnancy of unknown location: external validation of the hCG based M6NP and M4 prediction models in an emergency gynaecology unit.” in BMJ Open. We appreciate the time and effort that you have dedicated to providing valuable feedback on the manuscript. We have been able to incorporate changes to reflect most of your suggestions and the changes are highlighted within the manuscript.

Here is a point-by-point response to your comments and concerns (*responses in italic* and revised manuscript text yellow marked):

1. Please update the URL to www.earlypregnancycare.co.uk, the one currently mentioned does not work.

Author response: The former URL has been replaced.

2. Calibration intercept: I am not sure whether this is described or performed correctly. The text refers to 'the intercept of the curve', where the curve in the paper is based on loess. However, the calibration intercept refers to the intercept of the logistic model $\text{logit}(P(Y=1)) = a + LP$, i.e. a model where the slope has been fixed to 1 by including it as an offset. Is this how it is done?

However, after reconsideration and also according to reviewer's comment below, we find it more appropriate to allocate the group of patients with only one hCG sample as non-eligible (see answer below), regardless of whether imputation is an option.

The authors reply that 'the majority' of these cases were admitted due to 'the clinical course' and were therefore not eligible for a second hCG. What does that mean exactly? Related to this, more than half of these cases were EP/PPUL. So I wonder about the reason for not having a second hCG. E.g., perhaps many of these cases had their pregnancy confirmed on a follow-up ultrasound scan before taking the second hCG? I mean, if the pregnancy is confirmed on day 2, then it is not a PUL anymore and the model has no use. It would be good if the authors could provide details on the reason for not having a second hCG in the paper, and to provide descriptive statistics for these patients

Author response: As the reviewer points out, "if the pregnancy is confirmed on day 2, then it is not a PUL anymore". We have elaborated on the assessment of patients with one hCG, changed the flow chart in figure 1 and added a supplemental table on descriptive statistics for patients with 1 hCG as suggested by reviewer.

Revised text: A total of 1447 patients were screened for inclusion into the study. 239 patients had only one hCG sample, and 186 of those had their pregnancy located (134 EP and 46 IUP) before a second hCG was taken (Fig. 1). Another 33 patients did not have a second hCG since a miscarriage was suspected, although not confirmed, and urine pregnancy test was used as follow-up. Only 21 patients were lost to follow-up. All 239 patients with a missing second hCG were managed as non-eligible (Fig. 1). Descriptive statistics of these women are presented in supplemental table 2.

There were 1208 eligible women. However, 125 of these had a second hCG outside the targeted time frame. Although excluded from the validation cohort they were part of a sensitivity analysis (Fig. 1). Six women were diagnosed with a gestational trophoblastic disease and were not part of the validation cohort (Fig. 1). The validation cohort consisted of 1061 women. Descriptive statistics including hCG data of the validation cohort and women that were eligible but lost to follow-up (n=15) are presented in Table 1. (Lines 253-265)

4. Because of the previous issue, I think the description on line 52 and on line 314 is not complete. The authors mention complete case analysis as a limitation, but the sensitivity analysis only refers to patients where the second hCG is not in the targeted interval, not to patients with missing second hCG or that are lost to follow up. I would like these issues to be discussed as well.

Author response: Yes, it is true that the sensitivity analysis did not include patients with missing second hCG or those who were lost to follow-up. As for the 239 patients with a missing second value, we have explained under point 3, why a majority no longer could be regarded as PUL and were thus not eligible. As for the patients who were lost to follow-up, it is true that multiple imputation would be an option. However, given the small fraction of patients lost to follow-up, a sensitivity analysis using multiple imputation of those would produce virtually identical results as the current analysis.

We have added a clarification in the abstract (line 52): A limitation was the complete case analysis; however, a sensitivity analysis adding all women regardless of time interval between two hCG showed consistent performance for M6NP. (Lines 54-55)

In the Discussion (former line 314): We have added: We refrained from including patients lost to follow-up in the sensitivity analysis which could be a limitation of the study. However, given the small fraction of patients lost to follow-up, a sensitivity analysis using multiple imputation would produce virtually identical results as the current analysis. (Lines 346-349)

The latter text was partially added in the abstract (Lines 57-58).

5. Line 248 states that 89 were managed as inpatients. Why was that?

Author response: We have added the following explanation: The clinical picture warranted inpatient surveillance before a second hCG was taken for 8% of the women (89/1061). (Lines 273-275).

6. Line 270: there is no p-value for sensitivity, but there is one for FPR. Why?

Author response: A p-value has been added (line 297).

7. Line 273: “rendered 5% risk predictions” means “rendered risk estimates $\geq 5\%$ ”?

Author response: That is correct. We have changed the sentence: Less than 50% of failed PUL and 75% of IUP rendered risk estimates $\geq 5\%$ of an EP by M6NP. (Lines 300-301)

8. The cases with molar pregnancy should ideally not have been excluded. At the moment of making a prediction, it is not known that they are molar. I leave this up to the authors (we have been ‘guilty’ of this before until a reviewer pointed this out...).

Author response: Thank you for pointing this out to us as well. We agree and believe that the same consideration could be made for patients with a cancer diagnosis as well. We chose to add a short paragraph to the Discussion:

Six eligible women with a gestational trophoblastic disease were excluded. In a clinical situation a diagnosis is not known at the moment of prediction, why it might have been ideal to include these women. A post hoc analysis revealed that 5 out of 6 would have been classified as high risk, consequently affecting specificity negatively since being non-EP. Although a limited number, this must be taken into consideration when interpreting our data. (Lines 350-355)

9. Line 329: “M6NP mainly misclassified IUP and to a lesser extent failed PUL in our cohort as reported for M6P”, I do not understand this sentence. You cannot simply compare M6NP with M6P?

Author response: Also M6NP mainly misclassified IUP when validated in a UK study. We have changed the sentence and the reference:

M6NP mainly misclassified IUP and to a lesser extent failed PUL as EP in our cohort, consistent with findings in a UK validation study [14]. (Lines 364-365)

VERSION 3 – REVIEW

REVIEWER	Van Calster, Ben KU Leuven, Department of Development and Regeneration
REVIEW RETURNED	27-Jun-2022
GENERAL COMMENTS	We may have disagreements about missing values and imputation, but I will leave it at that.